# Effects of Various Processing Methods on the Metabolic Profile and Antioxidant Activity of *Dendrobium catenatum* Lindley Leaves

**DOI:** 10.3390/metabo11060351

**Published:** 2021-05-30

**Authors:** Jing-Jing Liu, Zhen-Peng Liu, Xin-Feng Zhang, Jin-Ping Si

**Affiliations:** State Key Laboratory of Subtropical Silviculture, Zhejiang A&F University, Hangzhou 311300, China; liujingjing@zafu.edu.cn (J.-J.L.); 18072846586@163.com (Z.-P.L.); zhangxf@zafu.edu.cn (X.-F.Z.)

**Keywords:** *Dendrobium catenatum* leaf, drying, LC-MS/MS, metabolite, antioxidant activity

## Abstract

The metabolite profiles and antioxidant activity of *Dendrobium catenatum* Lindley leaf, a new functional ingredient for food product development, were evaluated in samples that had been prepared using various methods, including freeze-drying, hot-air drying, rolling before drying, steaming before drying, steaming and rolling before drying, and drying at 100, 80, and 60 °C. The concentrations of polysaccharides and flavonoids, as well as the antioxidant capacity of each sample, were determined. Furthermore, two nucleosides, four amino acids, one monoaromatic compound, and eight flavonoids were identified in dried leaves using high-performance liquid chromatography–diode array detector–electrospray ionization–multistage mass spectrometry (HPLC-DAD-ESI-MS^n^) and ultraviolet (UV) spectral analyses. The content of polar compounds such as cytidylic acid, arginine, tyrosine, and hydroxybenzoic acid hexose increased dramatically during hot-air-drying and rolling-before-drying treatments, while flavonol *C*-glycosides remained stable throughout the various treatments and drying temperatures. Rolling before drying at 100 °C was identified as the most suitable process when manufacturing tea products from *D. catenatum* leaves. This process resulted in a high-antioxidant-activity and visually appealing tea. This report details a potential strategy that should be applied in the manufacturing processes of high-quality products from *D. catenatum* leaves.

## 1. Introduction

*Dendrobium catenatum* Lindley (synonym *D. officinale* Kimura et Migo) is one of the most valuable Chinese medicinal and edible herbs, and its dried stem is used as a crude drug named *Tiepi Shihu* [1]. It has been widely used as a tonic in China and other Southeast Asian countries for thousands of years. The importance of this species is highlighted by its high economic value, as the market price of wild *D. catenatum* is around $16,000 per kilogram [2,3]. However, wild *D. catenatum* is on the verge of extinction due to the destructive harvest, the destruction of the living environment, and its low reproduction ability. It has been listed in the *China Plant Red Data Book* and the list of second-class protected Chinese medicinal materials [4,5]. Fortunately, as the technology of variety selection and efficient farming developed, it has been cultivated on a large scale in recent decades. At present, the cultivation area of *D. catenatum* in China is more than 13,000 ha, with an output value over 2 billion dollars. It has become one of the fastest developing Chinese medicinal materials with the largest market share since the 1990s [5].

The *D. catenatum* leaf is a by-product of the stem, which comprises approximately half of the total biomass, and it is a new source of bioactive molecules that primarily include flavonoids, polysaccharides, amino acids, and other nourishing ingredients [6,7,8]. Moreover, the leaves exhibit potent antioxidant activity, and, in addition, antihyperlipidemia, antihypertensive, antihyperuricemia, anti-inflammatory, antimicrobial, cytotoxic and antitumor, hepatoprotective, immunomodulatory, lipase-inhibitory, and tyrosinase-inhibitory activities have been reported [8,9]. Therefore, *D. catenatum* leaves offer broad prospects for the development of new functional foods and were listed as a new raw food ingredient by the State Health Planning Commission of China in January 2017 [10]. Increasing interest in the nutritional and pharmacological properties of *D. catenatum* leaves motivated our investigation into their phytochemical composition, as well as the various manufacturing processes used to develop functional foods.

Dehydration is the primary technique used for the long-term storage of botanical products [11]. In the preparation of functional foods, freeze drying closely preserves flavor, color, and bioactive compounds. However, due to the high cost of freeze-drying techniques, hot-air-drying methods are commonly used [12]. Drying temperatures and treatment methods have been shown to substantially affect the yields of active ingredients and antioxidant capacity [13]. Our previous research regarding variations in the quality of *D. catenatum* stems that had been processed using distinct post-harvest methods revealed that twisting during drying at 80 °C is most effective for obtaining maximum polysaccharide yields [14]. These processing methods and chemical quality measurements were also used in the current study.

Chemical composition ultimately determines the taste and nutrient content of food. Various processing methods can affect the flavor and amounts of functional components. For example, freeze-dried red-fleshed apples have a higher anthocyanin content compared with apples that are thermally processed. Conversely, hot-air-dried products tend to contain higher concentrations of bioactive (poly)phenols [15]. The taste of green or black tea is influenced by the manufacturing process, which may include withering by blown air, steaming, rolling, and drying. Each of these processes can affect metabolite levels, such as water-soluble polysaccharides, catechins, and flavones, in the final tea product [16,17,18]. In a comparison of fresh *Cistanche deserticola* Ma samples with those that had been directly oven-dried and blanched, the levels of chemicals and bioactive ingredients were found to change throughout the steaming process [19]. The steaming of coffee beans can degrade methylxanthines and trigonelline, while increasing the concentrations of volatile compounds, nicotinic acid, and other derivatives [20]. In addition, in *Phyllanthus amarus* Shumacher & Thonning, the different drying methods (hot air, low-temperature air, infrared, microwave, sun, and vacuum), as well as the conditions within each method, substantially affect the phytochemical yield and antioxidant capacity [13]. 

Therefore, in the current study, we compared five distinct drying processes on the leaves of *D. catenatum*: freeze drying (FD), hot-air drying (D), rolling before drying (RD), steaming before drying (SD), and steaming and rolling before drying (SRD). Drying was performed at 100, 80, or 60 °C. Since flavonoids and polysaccharides are the main components in leaves [7,9,21], their content was determined by colorimetric method for each sample. In addition, changes in the scavenging rates of 2,2-diphenyl-1-picrylhydrazyl (DPPH) and 2,2′-azino-bis(3-ethylbenzothiazoline-6-sulfonic acid) diammonium salt (ABTS^+^) were measured. Furthermore, metabolite profiles were constructed using high-performance liquid chromatography (HPLC) fingerprints and HPLC–electrospray ionization (ESI)–multistage mass spectrometry (MS)^n^ methods combined with multivariate statistical analyses, including hierarchical cluster analysis (HCA) and partial least squares discriminate analysis (PLS-DA).

## 2. Results and Discussion

### 2.1. Leaf Morphology

Samples were treated as listed in Table 1. The morphologies of *D. catenatum* leaves that had been subjected to various manufacturing processes are shown in Figure 1. Freeze-drying (FD) leaves were bright green and flat and generally maintained their original shape (Figure 1A). Hot-air-drying (D) leaves were brown/green and flat and also generally maintained their shape (Figure 1B). Rolling-before-drying (RD) leaves were brown and formed into thin strips (Figure 1C). Steaming-before-drying (SD) and steaming-and-rolling-before-drying (SRD) leaves were darker (Figure 1D,E). No obvious morphological differences were observed as a function of drying temperature (60, 80, or 100 °C). The leaves were brewed into teas by soaking in hot water (approximately 85 °C) for 30 min up to 4 times. The teas were evaluated for appearance. The upper row of Figure 1F shows teas from leaves that had been hot-air-dried at 100 °C; the lower row displays teas from leaves that had been rolled and dried at 100 °C. Leaves that had been rolled yielded a more desirable tea featuring a pale gold appearance compared with tea made using hot-air-dried leaves. 

### 2.2. Antioxidant Activities of D. catenatum Leaves

The antioxidant activities of methanol-soluble extracts of *D. catenatum* leaves were evaluated using DPPH and ABTS assays (Figure 2A,B). Variation of drying temperature within an individual pretreatment method substantially influenced the scavenging capabilities of the extracts. Samples of D and RD leaves that had been dried at 100 °C showed the strongest antioxidant activity among all samples either in DPPH or in ABTS assay; leaves that had been freeze-dried maintained high levels of antioxidant activity; the decline in antioxidant capacity caused by drying temperature was obvious at 80 and 60 °C. In addition, at equivalent drying temperatures, steaming reduced antioxidant capacity, while rolling had no observable effects.

The antioxidant capacities of water-soluble extracts of *D. catenatum* leaves are shown in Figure 2C,D. No significant differences in DPPH or ABTS^+^ scavenging rates were observed among samples of D, RD, and SD leaves that had been dried at 100 or 80 °C. However, SRD samples exhibited reduced antioxidant activity. After drying at 60 °C, the antioxidant activity was lower in SD leaves than in RD leaves. Among all treatments, FD leaves exhibited the lowest antioxidant activity, presumably due to the low extraction efficiency of water.

### 2.3. Total Flavonoid and Water-Soluble Polysaccharide Content of D. catenatum Leaves

At equivalent drying temperatures, the levels of total flavonoid content decreased as follows: D > RD > SD > SRD (Figure 3A). Drying at 60 °C resulted in the lowest flavonoid yield. Drying D and RD leaves at 80 °C yielded more flavonoids compared with drying at 100 °C; for SD and SRD leaves, drying at 100 °C provided more flavonoids compared with drying at 80 °C; in addition, drying at 80 °C produced more flavonoids than drying at 60 °C. Pretreatments involving steaming presumably accelerated the deterioration of flavonoids. FD samples contained significantly (*p* < 0.05) greater amounts of total flavonoids compared with other samples. 

Steaming helped keep a high content of water-soluble polysaccharides in the leaves, while the drying temperature influenced a lot (Figure 3B). When drying directly or rolling before drying, 100 °C was the optimal temperature for polysaccharide preservation. Without steam processing, drying at 80 or 60 °C increased the recovery of water-soluble polysaccharides. FD and D leaves that had been direct-dried at 80 or 60 °C provided equal yields of water-soluble polysaccharides, which were lower than the yield obtained by drying at 100 °C. Rolling before drying at 60 °C yielded the lowest water-soluble polysaccharide content.

### 2.4. Correlations between Antioxidant Activity and Metabolite Content

Pearson correlation analysis showed a significant positive relationship between flavonoid content and the antioxidant activities of both DPPH· and ABTS^+^ scavenging rates in 80% methanol extract (0.445 and 0.829, both *p* < 0.01). However, a significant negative correlation was present between the water-soluble polysaccharide content and ABTS^+^-scavenging ability in the water extract (−0.330, *p* = 0.040). In addition, there was a negative correlation between flavonoid content and water-soluble polysaccharide content (−0.548, *p* = 0.000). Leaves that had been extracted with water showed no relationship between the water-soluble polysaccharide content and DPPH·-scavenging ability (*p* = 0.176), which was consistent with previous results that demonstrated no significant correlation between stem polysaccharide content and the DPPH· free radical clearance rate [22]. These results indicated that flavonoids are the major antioxidants in the leaves of *D. catenatum*, which is consistent with previous reports [9,21,22].

### 2.5. HPLC Fingerprinting and Similarity Analyses

Extracts of *D. catenatum* leaves contained abundant phenolic components. Following optimization of the separation conditions, more than 50 peaks were resolved in the HPLC chromatogram (Figure 4). Among these, 18 peaks corresponded to characteristic components (Figure 4A). Similarities among all samples were analyzed using the Similarity Evaluation System for Chromatographic Fingerprint of Traditional Chinese Medicine (Version 2004A). Samples D100, D80, D60, RD100, RD80, and RD60 exhibited low similarity (from 0.717 to 0.863); samples SD100, SD80, SD60, SRD100, SRD80, SRD60, and FD showed high similarity (>0.94, Appendix A). As shown in Figure 4B, the height of D and RD samples’ chromatographic peaks at 2~11 min was obviously higher than others, which indicated that different drying processes affect the content of macropolar metabolites. Notably, the metabolite profiles were similar for steamed samples and FD samples, while rolling had little effect on chemical composition. However, these similarity analyses were unable to effectively differentiate samples that had undergone different treatment processes.

### 2.6. HCA and PCA Results

HCA was an unsupervised classification procedure based on either distance or similarity between objects. In this study, one 26 × 18 matrix was formed using the relative areas of detected constituents from 13 fingerprint chromatograms that corresponded to the different samples and processing methods. Heat maps were generated using MeV v4.9.0 with Euclidean distancing to show the relationships between each of the 13 samples and the retention times of their chemical constituents. 

Three major HCA clusters were evident (Figure 5A). The first cluster was composed of D and RD samples that had been dried at 100 °C. The second cluster contained D and RD samples that had been dried at 80 °C and 60 °C. The third cluster contained all steamed samples and the FD samples. The high content of F1, F3, F6, and F7 (at retention times of 2.9, 3.4, 6.8, and 7.8 min, respectively) in clusters I and II clearly differentiated these samples from the other samples. 

The HCA heat map results were confirmed by principal component analysis (PCA) and PLS-DA. PCA was a non-parametric method of classification based on an orthogonal transformation to convert a set of possibly correlated variables into a set of linearly uncorrelated variables (i.e., principal components). Score plots of the two principal components (Figure 5B) showed significant differences between leaves according to their steam treatment status. In addition, FD leaves were similar to steamed samples. No obvious differences in metabolite content were observed among the three drying temperatures when samples were pretreated by steaming. However, there was difference between 100 °C and the lower temperatures (80 and 60 °C) when leaves processed by D and RD methods. These results were consistent with the HCA and HPLC fingerprint similarity analyses.

PLS-DA was used to visualize dissimilarities among the samples. Compared with PCA, PLS-DA was often preferred for sample discrimination because its dimension reduction was guided explicitly by variability among groups (FD, D, RD, SD, and RSD). For all drying methods, the first two principal components (45.6% and 43.1%) were considered statistically significant and constituted more than 88.7% of the total variance. Furthermore, three groups were easily differentiated in score plots (Figure 5C). In addition to the significant differences in metabolite composition that resulted from steaming, freeze-dried samples were also placed in a separate group. The intensity differences among metabolites with a variable importance in the projection value of >1.0 showed a robust ability to explain differences among groups. Peaks corresponding to F6, F2, F1, F17, F3, F5, F16, F7, and F4 were the most influential in discriminating among sample treatments. 

### 2.7. Identification of Metabolites by HPLC-DAD/ESI-MS^n^

Characterization of the chemical constituents of *D. catenatum* leaves was performed using an HPLC-DAD/ESI-MS^n^ system under optimal LC-ESI/MS^n^ conditions. Individual peaks were identified by comparison of retention times, UV absorbance spectra, mass spectra, parent ions, and fragment ions with available standards, references, and literature data. The UV absorbance and mass data are listed in Table 2.

The UV spectra of components corresponding to peaks F1 (*t_R_* = 2.88 min), F2 (*t_R_* = 3.20 min), and F5 (*t_R_* = 5.24 min) had similar spectral characteristics, indicating that the three compounds possess similar structural and functional groups. The spectra had a maximum absorption at 263–281 nm and additional absorption peaks at 215 or 223 nm, matching the UV absorption features of nucleosides [23]. The (+) ESI-MS spectra of peaks F1 and F2 produced parent ions at *m*/*z* 324.57 and 324.71, identifying them as cytidylic acid and uridylic acid, respectively. Peak F5 was inferred as a nucleoside analogue based on its UV absorption bands but was not identified via ESI-MS due to low ion abundance.

The UV spectrum of F3 (*t_R_* = 3.41 min) contained an absorption peak at 224 nm, indicative of an amino acid. (+) ESI-MS of F3 produced a parent ion at *m*/*z* 173.99, identifying it as arginine [24]. The UV spectra of F6 (*t_R_* = 6.78 min), F8 (*t_R_* = 9.64 min), and F9 (*t_R_* = 10.41 min) had similar spectral characteristics, with absorption peaks at 215–218 nm and 260–282 nm, corresponding to aromatic amino acids. (+) ESI-MS parent ions at *m*/*z* 182.44, 166.30, and 204.66 identified these compounds as tyrosine, phenylalanine, and tryptophan, respectively [24]. 

The UV absorbance spectrum of F7 (*t_R_* = 7.79 min) was consistent with a monoaromatic compound, including absorption bands at 214 and 298 nm. A molecular ion at *m*/*z* 301.00 [M+H]^+^ identified F7 as hydroxybenzoic acid hexose.

The UV absorbance spectra of flavonoids featured two major bands at 300–400 nm (band I) and 240–285 nm (band II). In negative or positive modes of ESI, the fragmentation pathway of flavonol *O*-glycosides (peak F18) indicated cleavage of the *O*-glycosidic bond, resulting in the loss of sugar moieties and retention of a charged aglycone fragment. In contrast, the negative ESI mode fragmentation pathway of flavonol *C*-glycosides (peaks F11 to F17) featured neutral losses of 120, 90, and/or 60 Da, resulting from cross-ring cleavages of hexose and/or pentose [25]. 

The deprotonated molecular ion peak of F18 (*t_R_* = 48.07 min) at (−) ESI-MS *m*/*z* 609.10 was assigned to a diglucoside, where fragmentation would result in the loss of a rhamnosyl moiety at *m*/*z* 463 and subsequent loss of a glucosyl moiety at *m*/*z* 301, retaining the aglycone fragment at *m*/*z* 301 [9]. Comparison with a standard compound yielded unambiguous identification as rutin.

The ESI-MS^n^ spectra and structures of seven flavonol *C*-glycosides from *D. catenatum* leaves are presented in Table 2 Peak F11, at 26.2 min, yielded an (−) ESI-MS [M-H]^−^ ion at *m*/*z* 593.08 (C_27_H_30_O_15_). This corresponded to vicenin II, consistent with mass transitions at *m*/*z* 353 [M-H-C_4_H_8_O_4_-C_4_H_8_O_4_]^−^, 383 [M-H-C_4_H_8_O_4_-C_3_H_6_O_3_]^−^, 473 [M-H-C_4_H_8_O_4_]^−^, and 503 [M-H-C_3_H_6_O_3_]^−^ to *m*/*z* 593 [M-H]^−^ [25,26]. Similarly, peaks F12, F13, and F14 produced an [M-H]^−^ ion at *m*/*z* 563 (C_26_H_28_O_14_), as well as a characteristic neutral loss of 90 Da, yielding a product ion at *m*/*z* 473; these findings corresponded to cross-ring cleavages at the 0 and 3 positions of a hexose group. The neutral losses of 60 Da (*m*/*z* 503) and 120 Da (*m*/*z* 443) corresponded to cross-ring cleavages at the 0 and 3 sites of a pentose group and 0 and 2 sites of a hexose group, respectively. Comparisons of relative abundance and retention times (elution order) with data from standard compounds identified F12, F13, and F14 as vicenin I, isoschaftoside, and schaftoside, respectively [25,26]. Similarly, peaks F15, F16, and F17 produced an [M-H]^−^ ion at *m*/*z* 533 (C_25_H_26_O_13_) and a characteristic neutral loss of 90 Da, yielding a product ion at *m*/*z* 443 from cross-ring cleavages at the 0 and 2 positions of a pentose group. The neutral loss of 60 Da (*m*/*z* 473) corresponded to cross-ring cleavages at the 0 and 3 sites of a pentose group. Compounds F15, F16, and F17 were tentatively identified as apigenin 6-*C*-*β*-D-xyloside-8-*C*-*α*-L-arabinoside, apigenin 6,8-di-*C*-*α*-L-arabinoside, and apigenin 6-*C*-*α*-L-arabinoside-8-*C*-*β*-D-xyloside, respectively, based on comparisons of relative abundances and retention times (elution order) with data from standard reference compounds [9,27,28,29].

## 3. Discussion

Among the various treatment methods, steaming reduced the antioxidant potential and the total flavonoid content but maintained the high level of polysaccharide yield. Rolling had no observable effects; however, the rolled leaf was easier to brew into a good-appearance tea, which may because the foliage cuticle and cell walls were destroyed during the rubbing process [14]. Furthermore, high-temperature pretreatment was conducive to form Maillard reaction products (MRPs), which markedly contribute not only to the aroma, taste and color but also to the antioxidant activity [30,31].

Freeze drying was generally regarded as an excellent method for maintaining the chemical components and biological activity of plant tissues, because the freezing and subsequent sublimation of water under vacuum conditions prevents the action of endogenous enzymes in leaves and changes in the metabolites of the samples [15,32]. In this study, the chemical composition of the steaming sample was most similar to that of the freeze-dried sample. This was because steaming is a good method to quickly destroy enzyme activity and is widely used in green tea manufacturing to eliminate the enzymatic oxidation process and obtain a relatively high content of active ingredients [33,34]. 

Apart from steaming, hot-air drying and rolling are also common methods of fixation [35]. Via high-temperature destruction and inactivation of oxidase activity in fresh leaves, fixation inhibited the enzymatic degradation of a variety of oxygen glycosides, including flavonoid glycosides and other components in fresh leaves; as some water evaporated from fresh leaves, it made the leaves soft and easy to roll, hence promoting the formation of a good aroma [35,36]. Both direct hot-air drying and rolling before drying at 100 °C achieved antioxidant activities similar to or better than the activities achieved by freeze drying. In contrast, low drying temperatures (80 and 60 °C) reduced the antioxidant activities of *D. catenatum* leaves. The antioxidant capacity of each group was closely related to the content of rutin (F18) and total flavonoids, which was consistent with our previous study that rutin has the strongest DPPH·/ABTS^+^ radical-scavenging activity among all the detected flavonoids in the leaves [9]. Similar to the manufacture of green, yellow, white, oolong, and black teas, fixation is crucial to maintain the stability of flavonol glycosides due to its inactivation effect on enzymes at high temperature [36]. 

Direct drying, as well as rolling before drying, dramatically increased the levels of polar molecules (F1–F7), including arginine (F3) and tyrosine (F6) (free amino acids), cytidylic acid (F1) and uridylic acid (F2) (nucleotides), and hydroxybenzoic acid hexose (F7). In these drying processes, because the fresh-leaf-withering process was relatively light, hydrolases were still active, so some proteins and peptides were hydrolyzed into amino acids under the catalysis of protease and peptidase; polysaccharides, cellulose, and other carbohydrates could continue to hydrolyze to form soluble sugars, which can coordinate the taste of tea and increase the precursors of aroma components; endogenous glycosidases catalyzed the degradation of glycosides, especially O-glycosides [37,38]. These products reacted with each other to form new substances, which was manifested by more peaks in the front part with retention times less than 11 min of the HPLC fingerprints. These similar metabolite changes have been observed with Pu-erh tea and *Angelicae Sinensis* Radix processes, presumably due to the enzymatic degradation process of macromolecular polymers [39,40].

Flavonoids were the major secondary metabolites in *D. catenatum* leaves, which eluted between 26 and 48 min in the HPLC fingerprints. Compared with freeze drying, steaming was most effective in preserving most of the original chemical compositions of the leaves. With the exception of F18 (rutin, a flavonol *O*-glycoside), most of the other flavonoids (flavonol *C*-glycosides) were insensitive to processing methods and temperatures, which was caused by the degradation effect of oxysidase on *O*-glycoside during processing and the stability of the *C*-glycoside bond [41]. 

It should be noted that Zhejiang and Yunnan Provinces are the major producing areas of *D. catenatum* in China, with a large artificial cultivation area. Tons of *D. catenatum* leaves are obtained in the harvest of the traditional medicinal tissue—the stem. The *D. catenatum* leaf is a new functional ingredient for food product, which has been developed in recent years. Therefore, an economical and efficient processing method is necessary to facilitate its storage and use. Compared with hot-air drying, the processing cost of lyophilizing the large number of leaves is high. According to our data, direct high-temperature drying is a relatively efficient method. Combined with rolling treatment such as green tea manufacture, it is conducive to local processing, storage, and transportation in each planting bases.

## 4. Materials and Methods

### 4.1. Preparation of Plant Materials

Leaves of *D. catenatum* were collected from the standard greenhouse of the Danxi Medicine-Expo Garden, Fotang Eco-agriculture Demonstration Zones, Yiwu, Zhejiang Province, China, as previously described [42,43,44]. The absolute location was east longitude, 120°04′; north latitude, 29°18′; and altitude, 100 m. *D. catenatum* was harvested on 6 February, when the stems were collected for commercial medicinal production. The leaves on biennial stems with a total fresh weight of 1990 g were collected from three test plots, each of which was 1 m^2^. 

### 4.2. Sample Preparation

Five manufacturing processes were evaluated: (1) fresh leaves were immediately freeze-dried (Christ alpha 1–2 LDplus; Marin Christ, Osterode, Germany); (2) fresh leaves were dried directly with hot air (air dry oven DGG-9070A; Senxin, Shanghai, China); (3) fresh leaves were rolled into thin strips before drying with hot air; (4) each batch of leaves was steamed for 5 min before drying; and (5) leaves were first steamed for 5 min and then rolled into thin strips before drying. Three drying temperatures (60, 80 and 100 °C) were evaluated for methods 2, 3, 4, and 5. After each sample had been dried to constant weight, it was crushed, passed through a 60-mesh sieve, and stored in a desiccator prior to use. 

### 4.3. Chemicals and Reagents

DPPH (≥98%) was purchased from Sigma-Aldrich (St. Louis, MO, USA), and ABTS^+^ (≥98%) was obtained from Solarbio (Beijing, China). Rutin, schaftoside, and isoschaftoside were purchased from the Chengdu Institute of Biology at the Chinese Academy of Sciences (Sichuan, China). All solvents used for plant extraction were of analytical grade and purchased from Sinopharm Chemical Reagent Co., Ltd. (Shanghai, China). HPLC- and LC-MS-grade acetonitrile (99.9%) was obtained from the Tedia Company (Fairfield, OH, USA).

### 4.4. Preparation of Test Sample Solutions for Total Flavonoid, Metabolite, and Antioxidant Capacity Analyses

The extraction method for methanol soluble components was slightly adjusted based on our reported literature [9]. A 0.4 g portion of each dried sample was extracted with 20 mL of 80% methanol in water in a 50 mL round-bottom flask in a 37 kHz ultrasonic water bath (Elma Elmasonic P120H; Elma Schmidbauer GmbH, Singen, Germany) at 20 °C for 30 min. The lost weight of the solution was implied after extraction, and the extracts were filtered through a 0.45 μm membrane filter prior to analyses for total flavonoid content, metabolite profiles, and antioxidant capacity.

### 4.5. Preparation of Test Samples for Water-Soluble Polysaccharide and Antioxidant Capacity Analyses

The water extraction and alcohol precipitation methods were used to extract polysaccharide solution, as previous reported [45,46]. A 0.2 g portion of each sample powder was placed into a 100 mL round-bottom flask. Distilled water (80 mL) was added, the mixture was heated under reflux at 100 °C for 2 h, and then cooled and filtered. The filtrate was transferred to a 100 mL volumetric flask, and the flask was filled to the mark with water. A 50 mL centrifuge tube was charged with 5 mL of the filtrate and 25 mL of absolute ethanol. The tube was shaken, refrigerated at 4 °C for 1 h, and then centrifuged at 6000 r·min^−1^ for 1 h. The supernatant was discarded, and 20 mL of 80% ethanol was added to the precipitate. This mixture was then centrifuged twice for 20 min. The resulting precipitate was dissolved with hot water in a 50 mL volumetric flask (filled to the mark with water). The flask was then shaken and allowed to cool to room temperature. The solution was used to analyze the water-soluble polysaccharide content and antioxidant capacity.

### 4.6. Determination of the DPPH Free-Radical-Scavenging Activities of Leaf Extracts

The scavenging of DPPH· free radicals was the basis of a common antioxidant assay [28,47]. In this work, the DPPH·-scavenging ability of each extract was measured, as described previously [9,28,48] with minor modifications. DPPH (7.89 mg) was dissolved in anhydrous 80% methanol and diluted to 100 mL to generate a stock solution (0.2 mmol·mL^−1^). Each sample solution (750 μL) was first mixed with 0.5 mL of the DPPH stock solution, then shaken vigorously, and incubated at room temperature in the dark for 40 min. The solution absorbance was measured at 517 nm using an ultraviolet–visible spectrophotometer (GENESYS 10S UV-VIS; Thermo Fisher Scientific, Waltham, MA, USA). All samples were analyzed in triplicate. Antioxidant activity was determined as follows:DPPH· free-radical-scavenging rate (%) = [1 − (*A*_sample_ − *A*_control_)/*A*_blank_] × 100,(1)
where *A*_blank_ was the absorbance of 750 μL of the solvent mixed with 0.5 mL of the DPPH working liquid, *A*_sample_ was the absorbance of 750 μL of the sample mixed with 0.5 mL of the DPPH working liquid, and *A*_control_ was the absorbance of 750 μL of the sample mixed with 0.5 mL of the solvent. 

### 4.7. Determination of the ABTS^+^ Free-Radical-Scavenging Activities of Leaf Extracts

The ABTS^+^ radical-scavenging capacity of each extract was measured using a common antioxidant assay [9,49,50], with minor modifications. Solutions were prepared with 5 mL of ABTS^+^ solution (7 mmol·L^−1^) and 1 mL of aqueous potassium persulfate (140 mmol·L^−1^). Then, 88 μL of potassium persulfate solution was mixed with 5 mL of ABTS^+^ solution and incubated in the dark at room temperature for 16 h to produce a stock solution of ABTS^+^. Working solutions of ABTS^+^ were prepared by dilution in absolute ethanol to an absorbance of 0.7 (±0.02) at 734 nm. The working solution (975 μL) of ABTS^+^ was mixed with 25 μL of each sample test solution. All mixtures were shaken for 30 s, incubated in the dark at room temperature for 6 min, then subjected to absorbance measurement at 734 nm. All experiments were performed in triplicate. The percentage reduction of ABTS^+^ by each dilution was determined as follows:ABTS^+^ free-radical-scavenging rate (%) = [(*A*_blank_ − *A*_sample_)/*A*_blank_] × 100,(2)
where *A*_blank_ was the absorbance of 25 μL of the solvent mixed with 975 μL of the ABTS^+^ working solution and *A*_sample_ was the absorbance of 25 μL of the sample mixed with 975 μL of the ABTS^+^ working solution.

### 4.8. Determination of Total Flavonoid Content

The total flavonoid content of the leaves was measured using a modified NaNO_2_-Al(NO_3_)_3_-NaOH colorimetric method, based on the formation of a flavonoid–aluminum complex with a maximum absorbance at 510 nm [28]. Briefly, 2 mL of leaf extract was placed in a 25 mL volumetric flask and mixed with 1.0 mL of 5% sodium nitrite solution, 1.0 mL of 10% aluminum nitrate, and 10.0 mL of 4% sodium hydroxide at respective intervals of 6 min. Finally, the flask was filled to the mark with 80% methanol. The absorbance of the mixture was immediately measured at a wavelength of 510 nm against a prepared blank using a UV–VIS spectrophotometer. The flavonoid content was determined by comparison with a rutin standard curve and stated as the mean (mg rutin equivalents per g leaves) ± standard deviation of triplicate experiments. The standard regression equation of rutin was *y* = 9.3774*x* − 0.0018, *R^2^* = 0.9998; *y* represents the solution absorbance, and *x* represents the concentration of rutin.

### 4.9. Determination of the Water-Soluble Polysaccharide Content

The concentrations of water-soluble polysaccharides were determined using a phenol–sulfuric acid method [46,51] as follows: a test tube containing 1 mL of the sample was placed in ice water. To this chilled test tube was added 1 mL of 5% phenol solution and 5 mL of concentrated sulfuric acid. The mixture was shaken, and the test tubes were placed in ice water for 5 min, then boiled in hot water for 20 min, and then cooled to room temperature. The absorbance of each solution was then measured at 488 nm. Distilled water was used as the blank in a set of three parallel experiments.

A glucose standard curve was prepared as a reference. Anhydrous *D*-glucose (10 mg) was dissolved in 100 mL of water and then diluted to 0.01, 0.015, 0.02, 0.04, 0.06, and 0.1 mg·mL^−1^. According to the phenol–sulfuric acid method referenced above, the absorbance of each glucose concentration was used to generate a standard curve of absorbance as a function of concentration. Polysaccharide content was determined by comparing the sample absorbance against the *D*-glucose standard curve and was stated as the mean (mg *D*-glucose equivalents per g leaves) ± standard deviation of triplicate experiments. The regression equation of the standard glucose curve was *y* = 9.3911*x* − 0.0161, *R^2^* = 0.9999; *y* represents the solution absorbance, and *x* represents the concentration of *D*-glucose.

### 4.10. HPLC Fingerprinting

Qualitative and quantitative HPLC analyses were performed on an Agilent 1200 system HPLC equipped with a reverse-phase Venusil MP C_18_ column (4.6 mm × 250 mm, 5 μm), a diode array detector (DAD), a quaternary pump, a column temperature box, and an automatic injector. The mobile phases were composed of aqueous 0.5% acetic acid (A) and acetonitrile (B). Elution was performed in accordance with the following linear gradient: From 0 to 18 min, solvent B was increased from 5% to 17% and then maintained at 17% until 33 min. From 33 to 43 min, solvent B was increased from 17% to 21%; from 43 to 48 min, it was increased from 21% to 23%. From 48 to 50 min, solvent B was decreased from 23% to 5% and then maintained at 5% until 60 min. The flow rate was maintained at 1 mL·min^−1^. The column temperature was controlled at 30 °C. The injection volume was 10 μL, and the detection wavelength was 270 nm [9]. 

### 4.11. Data Processing and Statistical Analyses

HPLC data were first processed by Agilent ChemStation software and exported as AIA files to the Similarity Evaluation System for Chromatographic Fingerprint of Traditional Chinese Medicine version 2004A (Chinese Pharmacopoeia Commission, Beijing, China) for calibration of peak retention times and similarity analyses. The normalized HPLC data were imported into MeV v4.9.0 for hierarchical clustering analyses [52]. Classification was performed focusing on the rows and columns. The Euclidian distance method was used to measure the proximity between samples and peaks; results were visualized as dendrograms on two-dimensional heat maps. PCA and PLS-DA were performed using SIMCA-P V11.0 software (Umetrics, Umea, Sweden). Variables that significantly contributed to the classification were identified according to a threshold of variable importance in the projection (VIP) values, as generated by PLS-DA. Statistical analyses were performed with one-way analysis of variance using SPSS Statistics version 17.0 (SPSS Inc., Chicago, IL, USA). Differences with *p* < 0.05 were considered statistically significant.

### 4.12. Identification of Metabolites by HPLC-DAD/ESI-MS^n^ and HR-ESI-MS

Herb markers corresponding to chromatographic peaks from HPLC analyses of leaf extracts of *D. catenatum* were identified by HPLC-ESI-MS^n^ analyses and by comparisons of metabolite fragmentation patterns with the patterns of standard reference materials. HPLC-ESI-MS^n^ was performed on a Thermo Finnigan LCQ Deca XP Electrospray Ion Trap Mass Spectrometer (ThermoQuest-Finnigan Co., San Jose, CA, USA) equipped with an ESI source using helium as the collision gas for MS/MS experiments. Multiple scanning modes were cyclically alternated during the analyses in a data-dependent manner, in accordance with our existing process [48] with minor modifications: (1) primary scans were performed in full scan mode ranging from *m*/*z* 100–2000 Da, and (2) secondary scans were data-dependent MS/MS with a relative collision energy of 20–35% and wideband activation features were enabled. The isolation width was set to 1 Da, and ejected ions were detected with an electron multiplier at a gain of 5 × 10^5^. Data acquisition and processing were performed with Xcalibur Software version 1.4 (Thermo Fisher Scientific). For HPLC–ultraviolet (UV)/ESI-MS^n^ analyses, the HPLC eluent was subjected to a 1:2 split before introduction to the ion source from the mass spectrometer. High-resolution (HR)-ESI-MS detection was performed on a Agilent 6550 iFunnel/Q-TOF mass spectrometer with an Agilent Jet-Stream source. The ESI source was operated in positive and negative ionization modes with a capillary voltage of 3.5 kV for both modes, a nozzle voltage of 250 V (+) and 1500 V (−), a fragmentor voltage of 380 V, the nebulizer at 25 psi, and the sheath gas and dry gas set at a flow rate of 12 and 16 L/min, respectively.

## 5. Conclusions

Metabolite and antioxidant activity differences among *D. catenatum* leaves via five different drying processes and three different temperatures were explored. The results showed that lyophilization and steaming can be used to rapidly process plant materials, while preserving most of the original compounds; drying at a high temperature (100 °C) without rolling and steaming helps preserve the antioxidative properties and most of the leaf metabolites; hot-air drying and rolling before drying increase the nucleoside and amino acid content. In conclusion, rolling before drying at 100 °C, which preserved the chemical profiles of the original materials, increased the taste and flavor of large polar components, and maintained high antioxidant levels, was identified as a suitable manufacturing process for *D. catenatum* leaves that will be used in tea. Future studies and industrial application aiming at manufacture of *D. catenatum* leaf and other herbal teas could take advantage of this work to choose an appropriate drying method and evaluate product quality. 

## Figures and Tables

**Figure 1 metabolites-11-00351-f001:**
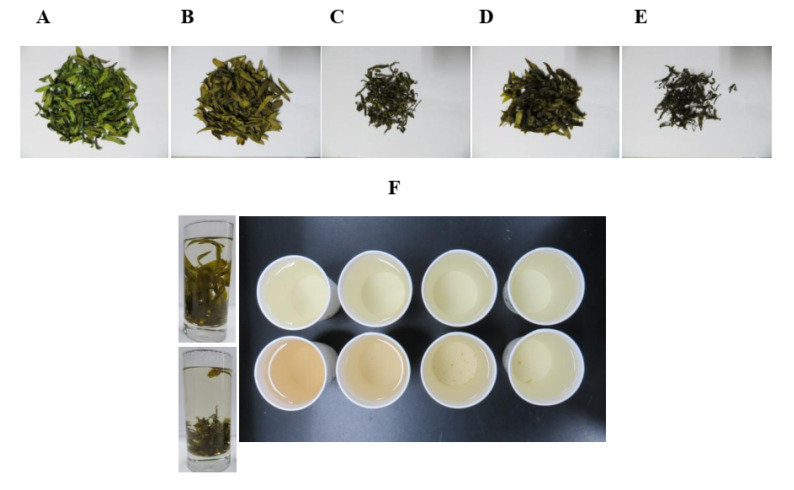
Morphologies of *Dendrobium catenatum* leaves are shown after exposure to various manufacturing processes: (**A**) freeze drying, (**B**) hot-air drying at 80 °C, (**C**) rolling before drying at 80 °C, (**D**) steaming before drying at 80 °C, and (**E**) steaming and rolling before drying at 80 °C. (**F**) Teas were brewed in hot water (85 ± 5 °C). The upper panel show teas made from leaves dried with hot air at 100 °C; the lower panel shows teas made with leaves dried at 100 °C after rolling.

**Figure 2 metabolites-11-00351-f002:**
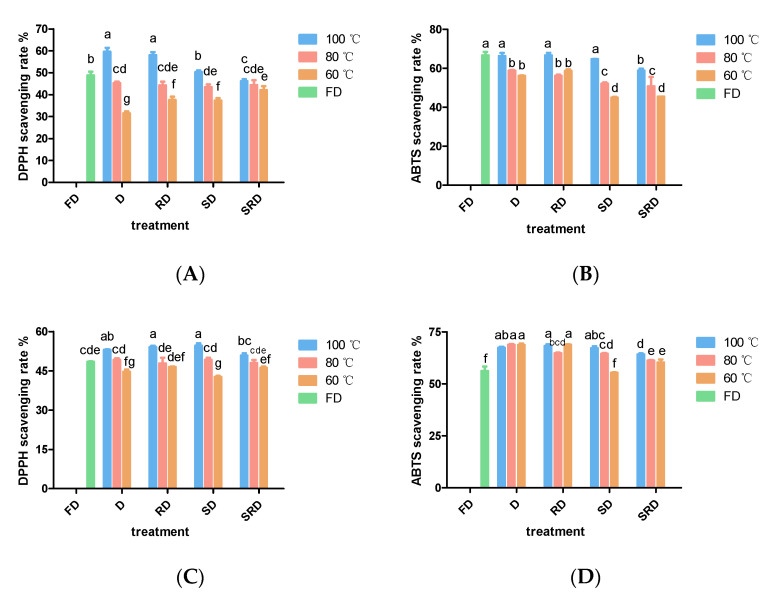
Antioxidant activities of *Dendrobium catenatum* (0.02 g·mL^−1^ dried weight) are shown after exposure to various processing methods: (**A**) DPPH·-scavenging rate of 80% methanol extract, (**B**) ABTS^+^-scavenging rate of 80% methanol extract, (**C**) DPPH·-scavenging rate of water extract, and (**D**) ABTS^+^-scavenging rate of water extract. Columns represent the mean (*n* = 3) ± standard deviation. Columns of the same shape with different letters indicate significant differences (*p* < 0.05).

**Figure 3 metabolites-11-00351-f003:**
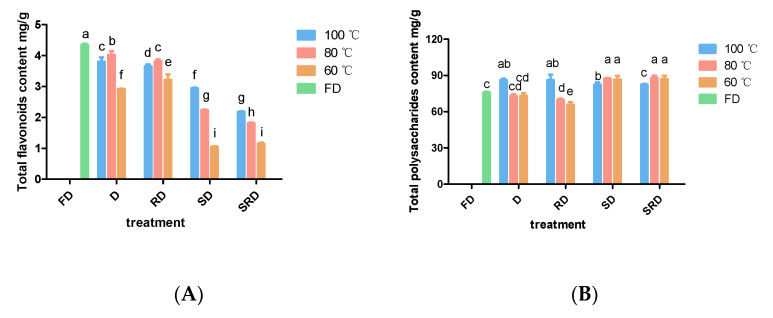
Total flavonoid and polysaccharide contents of *Dendrobium catenatum* leaves are shown after exposure to various processing methods: (**A**) total flavonoid content and (**B**) total polysaccharide content. Columns represent the mean (*n* = 3) ± standard deviation. Columns of the same shape with different letters indicate significant differences (*p* < 0.05).

**Figure 4 metabolites-11-00351-f004:**
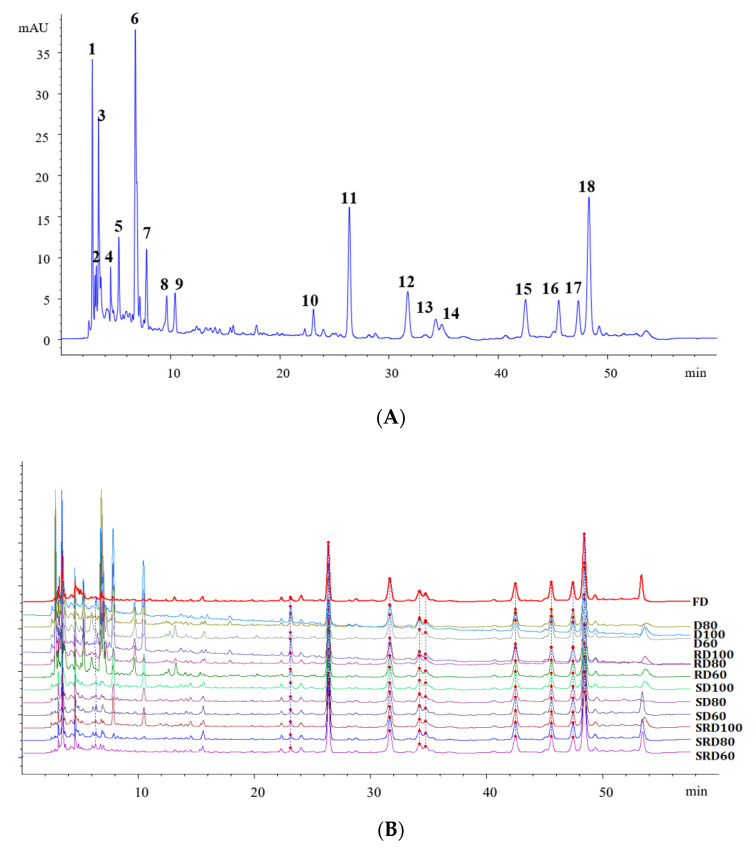
Fingerprinting analyses of *Dendrobium catenatum* leaves: (**A**) representative HPLC chromatogram (**B**) and chromatograms obtained after exposure to various processing methods.

**Figure 5 metabolites-11-00351-f005:**
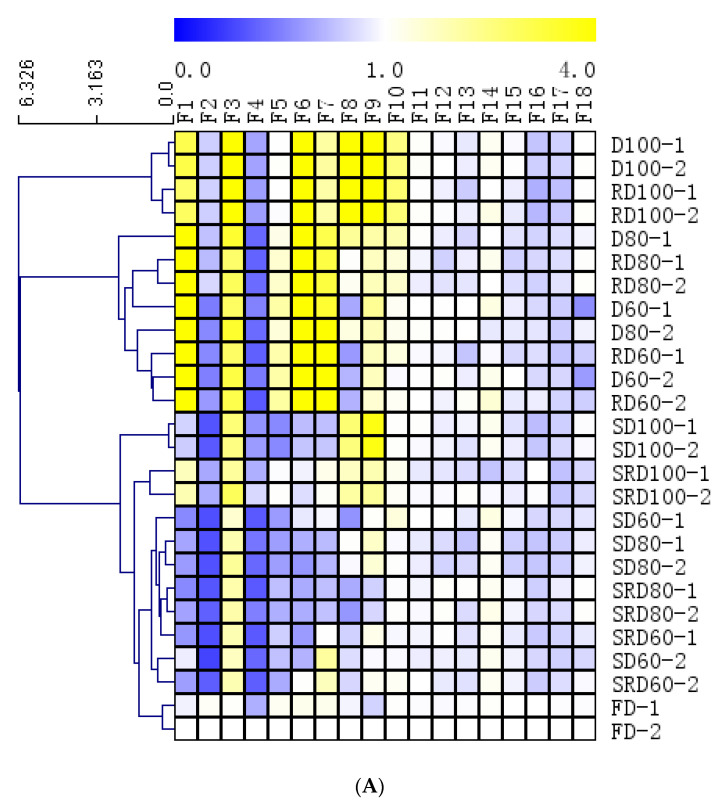
(**A**) Heat map depicting the results of hierarchical cluster analysis using the Euclidean distance method, and (**B**) PCA score plots and (**C**) PLS-DA data obtained from *Dendrobium catenatum* leaves after exposure to various processing methods.

**Table 1 metabolites-11-00351-t001:** Manufacturing processes used for *Dendrobium catenatum* leaves.

Treatments	Abbreviation	Drying Temperature		
Freeze drying	FD	−60 °C		
Hot-air drying	D	60 °C	80 °C	100 °C
Rolling before drying	RD	60 °C	80 °C	100 °C
Steaming before drying	SD	60 °C	80 °C	100 °C
Steaming and rolling before drying	SRD	60 °C	80 °C	100 °C

**Table 2 metabolites-11-00351-t002:** Metabolites from *Dendrobium catenatum* leaves identified by HPLC-DAD/ESI-MS^n^.

Peak.	Rt (min)	λ max (nm)	Compound Names	Molecular Weight (Da)	Error (ppm)	ESI-MS [M+H]^+^ (m/z)	ESI-MS [M-H]^−^ (m/z)	Negative ESI-MS/MS (Relative Abundance, %)	Type
F1	2.88	223, 281	Cytidylic acid	323.0524	−1.76	324.57	-	-	Nucleoside
F2	3.20	223, 275	Uridylic acid	324.0355	1.05	324.71	-	-	Nucleoside
F3	3.41	224	Arginine	174.1105	6.69	173.99	-	-	Amino acid
F4	4.53	-	Unknown	-	-	-	-	-	Unknown
F5	5.24	215, 263	Unknown	-	-	-	-	-	Nucleoside analogue
F6	6.78	218, 260	Tyrosine	181.0738	0.30	182.44	-	-	Amino acid
F7	7.79	215, 298	Hydroxybenzoic acid hexose	300.0844	0.53	301.00	-	-	Monoaromatic
F8	9.64	217, 265	Phenylalanine	165.0788	1.00	166.30	-	-	Amino acid
F9	10.41	218, 282	Tryptophan	204.0914	−7.42	204.66	-	-	Amino acid
F10	22.97	296	Unknown	-	-	-	-	-	Unknown
F11	26.20	272, 322	Vicenin II	594.1620	−0.80	594.78	593.08	593 (47), 575 (9), 503 (32), 473 (100), 455 (5), 383 (11), 353 (13)	Flavonoid C-glycoside
F12	31.45	272, 322	Vicenin I	564.1504	4.38	564.78	563.07	563 (4), 545 (30), 503 (73), 473 (100), 443 (67), 425 (7), 383 (22), 353 (30)	Flavonoid C-glycoside
F13	33.98	272, 322	Isoschaftoside	564.1507	5.03	564.83	563.08	563 (21), 545 (17), 503 (82), 473 (100), 443 (92), 425 (11), 383 (33), 353 (25)	Flavonoid C-glycoside
F14	34.51	272, 322	Schaftoside	564.1479	−1.78	564.91	563.11	563 (11), 545 (14), 503 (7), 473 (48), 443 (100), 425 (2), 413 (2), 383 (7), 353 (16)	Flavonoid C-glycoside
F15	42.18	272, 322	Apigenin 6-C-β-D-xyloside-8-C-α-L-arabinoside	534.1407	6.29	534.75	533.02	533 (57), 473 (19), 443 (100), 383 (13)	Flavonoid C-glycoside
F16	45.26	272, 322	Apigenin 6, 8-di-C-α-L-arabinoside	534.1381	6.51	534.80	533.18	533 (12), 515 (18), 503 (2), 473 (45), 443 (100), 425 (5), 413 (3), 383 (7), 353 (5)	Flavonoid C-glycoside
F17	47.10	272, 322	Apigenin 6-C-α-L-arabinoside-8-C-β-D-xyloside	534.1398	4.68	534.85	533.15	533 (20), 515 (9), 503 (2), 473 (74), 443 (100), 425 (5), 383 (19), 353 (11)	Flavonoid C-glycoside
F18	48.07	257, 352	Rutin	610.1562	−5.96	610.57	609.10	609 (30), 591 (4), 463 (2), 373 (3), 343 (15), 301 (100), 271 (10)	Flavonoid O-glycoside

## Data Availability

All of the reported data is included in the manuscript.

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
