# Peer review of "Effects of Various Processing Methods on the Metabolic Profile and Antioxidant Activity of Dendrobium catenatum Lindley Leaves"

_metabolites, 2021, doi:10.3390/metabo11060351_

Round 1

Reviewer 1 Report

Comments and suggestion for Authors are in the attached file.

Author Response

The manuscript entitled ‘Effects of Various Processing Methods on the Metabolite Profiles and Antioxidant Capacity of Dendrobium catenatum Leaves’ is a very interesting paper about the comparison of different techniques for preserving metabolites in these leaves, widely used for teas. It is well performed, and, in my opinion, fits perfectly with the journal scope, but this is up to the editor to decide. However, at present, the work cannot be published, as it lacks important information. Therefore, I suggest some following major revisions:

An extensive English editing is required.

Answer: Thank you very much for your comments and suggestion. The manuscript has been checked and corrected the English by two editors. A specialist editor with suitable professional knowledge reviewed and corrected the English. An English language specialist subsequently checked the paper again. The first language of both editors is English. The certify could be downloaded in the following website. http://www.textcheck.com/certificate/d2f7ue

-INTRODUCTION. The first part of introduction appears as a list of information with low consequentiality between the sentences. For example, at line 32: please, you may introduce the prize of wild Dendrobium with some fitting phrase as ‘the importance of this specie is highlighted by its high economic value, ranging around $16,000…’; introduce the advantages and benefits from its cultivation and biomass.

Answer: Line 32: The sentence has been modified to “The importance of this specie is highlighted by its high economic value, as the market price of wild D. catenatum is around $16,000 per kilogram”. In addition, the introduction has been revised to enrich the information density.

Line 68: please rephrase to give more continuity to the text, e.g. ‘Also in Phyllanthus amarus, the different drying methods (hot air, 66 low-temperature air, infrared, microwave, sun, and vacuum), as well as the conditions 67 within each method, substantially affect the phytochemical yield and antioxidant capacity’.

Answer: The sentence has been modified as sugested: “Also in Phyllanthus amarus Shumacher & Thonning, the different drying methods (hot air, low-temperature air, infrared, microwave, sun, and vacuum), as well as the conditions within each method, substantially affect the phytochemical yield and antioxidant capacity”.

Line 70: please rephrase to give more continuity to the text, e.g. Therefore, in the current study, we have compared five distinct drying processes on the leaves of D. catenatum.

Answer: The sentence has been modified as sugested: “Therefore, in the current study, we have compared five distinct drying processes on the leaves of D. catenatum”.

-RESULTS AND DISCUSSION:

Please, add the full name of acronyms when mentioned for the first time (FD, D, RD, SD, SRD).

Answer: The full names have been added: freeze drying (FD), hot air drying (D), rolling before drying (RD), steaming before drying (SD), and steaming and rolling before drying (SRD).

Line 93: Authors stated that teas from rolled leaves yielded a more desirable tea featuring a pale gold appearance and a better taste, compared with tea made using hot air-dried leaves. However, it should be interesting to know how was the analysis conducted? How many people expressed this judgment? Was it a blind study?

Answer: That's a very good question. The rolled leaves yielded a more desirable tea featuring a pale gold appearance and a better taste was the result of our observation and tasting of the D. catenatum leaf tea after processing (Figure1 F). Unfortunately, there was no scientific, systematic and data-based evaluation and statistical analysis.

Figure 2 appears with a white line in the graphs. Please, see comments of material and methods and express data as IC50 or mg standard equivalents/grams of extract.

Answer: The graphs in Figure 2 have been modified to eliminate the white line.

Indeed, you suggested using IC50 or mg standard equivalents/grams of extract is a better way to describe the antioxidant activity. However,  it is also a feasible and simple method to compare the radical scavenging rate of different samples by using the fixed extract concentration as we described in previously published paper: Zhang, Y.; Zhang, L.; Liu, J.; Liang, J.; Si, J.; Wu, S. Dendrobium officinale leaves as a new antioxidant source. Journal of Functional Foods 2017, 37, 400–415.

It was described in the original text: As showed in Fig. 2A, the methanol extracts of leaves of D. officinale showed significant radical scavenging capacity in DPPH assay.In contrast with two well-known antioxidant standard vitamin C (Vc, 50 g· mL-1) and butylated hydroxytoluene (BHT, 50 g· mL-1), 20 g· mL-1 dried leaves sample solution also showed approximate antioxidant activities. The radical scavenging rate of the sample of strain 86×56 collected in February was significantly higher than those of other strains.

The raw materials used in this revised paper was 6A×2B strain, the same strain with similar antioxidant capacity in Fig.2. But different processing methods do have significant effects on the composition and antioxidant capacity. Therefore, the comparison method of radical scavenging rate under fixed concentration was retained in the revised draft.

In general, results should be more discussed, taking into account previous data from literature. For example, when you state ‘Leaves that had been extracted with water showed no significant relationship between water-soluble polysaccharide content and DPPH scavenging ability (P = 154 0.176)’ or ‘Theoretically, no polysaccharides were extracted with 80% methanol’. Are there articles in literature with same or different outcomes?

Answer: Thank you for your suggestion. The result “Leaves that had been extracted with water showed no significant relationship between water-soluble polysaccharide content and DPPH scavenging ability” was was similar with the result that there was no significant correlation between antioxidant activity and clearance rate of stem polysaccharide extracts [21]. In addition, the result “flavonoids were the major antioxidants in the leaves of D. catenatum” was consistent with previous reports [9,21,22]. These references have been added in the revised paper.

Also for metabolite identification. Are the identified compounds the same found in literature?

Answer: No new compounds have been identified in this study, so all the identified compounds were the same found in literature.

-MATERIAL AND METHODS:

Please rephrase line 322.

0.4-g should be 0.4 g.

Answer: It has been modified as sugested.

Why the samples have been extracted in two different ways (solvent and techniques) in sections 4.4 and 4.5? Please motivate your choice, adding appropriate reference.

Answer: In this paper, two different extraction methods were used to extract methanol soluble components and water soluble components especially the polysaccharide, in order to investigate compounds with different polarity. The references have been added.

As regards antioxidant activity, it should be better express the activity in a more univocal way, as well as IC50 or mg standard equivalents/grams of extract, basing on a calibration curve of a standard. Please, re-elaborate the data.

Answer:  Thank you for your suggestion. Please find the answer in the revision notes to Figure 2.

Why did you focus only on flavonoids? The determination of total phenolic content by Folin-Ciocalteau method should improve the quality of manuscript.

Answer: According to the literature, the main component of D. candidum leaves is flavonoid, so we focused on the content of flavonoids. e.g

Zhang, Y.; Zhang, L.; Liu, J.; Liang, J.; Si, J.; Wu, S. Dendrobium officinale leaves as a new antioxidant source. Journal of Functional Foods 2017, 37, 400–415.

Zhou, G.; Lv, G. Comparative studies on scavenging DPPH free radicals activity of flavone C-glycosides from different parts of Dendrobium officinale. China journal of Chinese materia medica 2012, 37, 1536-1540.

CONCLUSIONS should be ampliated with further employments in nutraceutical or food sector.

Answer: The conculsions have been revised and this sentence has been added: “Future studies and industrial application aiming at manutucture of D. catenatum leaf and other herbal tea could take advantage of this work to choose appropriate drying method and evaluate product quality.”

Line 1: ‘the metabolic profile and antioxidant activity of Dendrobium catenatum leaves’ should be more correct. Please, add the botanical classifier.

Answer: It has been modified as sugested. The botanical classifier has been added.

Line 17: analysis; the content of.. (please, verify within the text ‘content of plurals’ and not ‘contents’ e.g. line 74)

Answer: The word content have been modified as sugested. “Analyses” is the plural form of “analysis” and kept in the original manuscripts.

Line 21: should be used…

Answer: The word has been modified as sugested.

Line 24: potential strategy that should be applied in the manufacturing processes of high-quality products from D. catenatum leaves.

Answer: The sentence has been modified as sugested.

Line 29: use period instead of comma before ‘the dried steam of leaf….’

Answer: It has been modified as sugested.

Line 32, 37, etc.: check the spaces before the references.

Answer: It has been modified as sugested.

Line 61-68: add botanical classifier to Cistanche deserticola and Phyllanthus amarus.

Answer: The botanical classifiers have been added.

Check the tenses: line 61, 90: that have been

Answer: They have been modified as sugested.

Line 91 the lower row not rows

Answer: It has been modified as sugested.

Caption of figure 1: panel not panels

Answer: It has been modified as sugested.

Check the bibliography: for example, reference 9 does not show volume and issue

Answer: The volume and page has been added.

Reviewer 2 Report

Generally, the paper have good experimental work but although the paper is well structured and the methodology correctly developed, the paper cannot be considered for publication in its actual format. 

You need to review English. I have pointed out some of them in the comments, but a more careful revision is suggested. 

Lines 32-34: I suggest to modify as following: 

“As the technology of variety selection and efficient farming has been developed, it has been cultivated on a large scale in recent decades 

Lines 34-36: I suggest to modify as following: 

The D. catenatum leaf comprises approximately half of the total biomass and it is a new source of bioactive molecules such as phenols, amino acids and other nourishing ingredients, above all as tea. 

Lines 45-46: I suggest to modify as following: 

However, due to the high cost of freeze-drying techniques, hot air-drying methods are commonly used. 

Line 83, 112: “were” instead of “are 

Line 90: showed” instead of “shows 

Line 91: displayed” instead of “show” 

Line 129: “provided” instead of “yielded” 

Line 130: “produced” instead of “yielded” 

Line 156 

Lines 179-180: I suggest to modify as following: 

HCA was an unsupervised classification procedure based on either distance or similarity between objects.” 

Lines 192-194: I suggest to modify as following: 

PCA was a non-parametric method of classification based on an orthogonal transformation to convert a set of possibly correlated variables into a set of linearly uncorrelated variables (i.e., principal components).” 

Line 202 and 203: “was” instead of “is 

Line 225: “were” instead of “are” 

Line 229: “possessed” instead of “possess” 

Line 229: “had” instead of “have 

Line 231: references must be substituted by number in square brackets 

Line 246: “featured” instead of “feature” 

Line 259: “were” instead of “are” 

Lines 313-314: how many plants did you use for the experiment? and how many leaves for each plant have you collected? Please specify 

Lines 334-339: is this a method developed in your lab? If not, please add reference 

Lines 342-351: is this a method developed in your lab? If not, please add reference 

Is the antioxidant activity evaluated both in the extract prepared for the flavonoids and for the polysaccharides? Please check 

Lines 397-412: is this a method developed in your lab? If not, please add reference 

Line 287: “was” instead of “is” 

Lines 279-307: I suggest you broaden the discussion. You did a great job with so many experiments and results but the discussion is really small compared to the rest of the paper. 

Author Response

Generally, the paper have good experimental work but although the paper is well structured and the methodology correctly developed, the paper cannot be considered for publication in its actual format. 

You need to review English. I have pointed out some of them in the comments, but a more careful revision is suggested. 

Answer: Thank you very much for your comments and suggestion. The manuscript has been checked and corrected the English by two editors. A specialist editor with suitable professional knowledge reviewed and corrected the English. An English language specialist subsequently checked the paper again. The first language of both editors is English. The certifycould be downloaded in the following website. http://www.textcheck.com/certificate/d2f7ue

Lines 32-34: I suggest to modify as following: 

“As the technology of variety selection and efficient farming has been developed, it has been cultivated on a large scale in recent decades” 

Answer: The sentence has been modified as sugested.

Lines 34-36: I suggest to modify as following: 

“The D. catenatum leaf comprises approximately half of the total biomass and it is a new source of bioactive molecules such as phenols, amino acids and other nourishing ingredients, above all as tea.” 

Answer: The sentence has been modified as sugested.

Lines 45-46: I suggest to modify as following: 

“However, due to the high cost of freeze-drying techniques, hot air-drying methods are commonly used”. 

Answer: The sentence has been modified as sugested.

Line 83, 112: “were” instead of “are” 

Answer: The words have been modified as sugested.

Line 90: “showed” instead of “shows” 

Answer: The word has been modified as sugested.

Line 91: “displayed” instead of “show”  

Answer: The word has been modified as sugested.

Line 129: “provided” instead of “yielded”  

Answer: The word has been modified as sugested.

Line 130: “produced” instead of “yielded”  

Answer: The word has been modified as sugested.

Line 156:  

Answer: The sentence has been modified-- “were” instead of “are”.

 Lines 179-180: I suggest to modify as following: 

“HCA was an unsupervised classification procedure based on either distance or similarity between objects.” 

Answer: The sentence has been modified as sugested.

Lines 192-194: I suggest to modify as following: 

“PCA was a non-parametric method of classification based on an orthogonal transformation to convert a set of possibly correlated variables into a set of linearly uncorrelated variables (i.e., principal components).” 

Answer: The sentence has been modified as sugested.

Line 202 and 203: “was” instead of “is” 

Answer: The words have been modified as sugested.

Line 225: “were” instead of “are” 

Answer: The word has been modified as sugested.

Line 229: “possessed” instead of “possess” 

Answer: The word has been modified as sugested.

Line 229: “had” instead of “have” 

Answer: The word has been modified as sugested.

Line 231: references must be substituted by number in square brackets 

 Answer: The reference number has been added.

Line 246: “featured” instead of “feature” 

Answer: The word has been modified as sugested.

Line 259: “were” instead of “are” 

Answer: The word has been modified as sugested.

Lines 313-314: how many plants did you use for the experiment? and how many leaves for each plant have you collected? Please specify 

Answer: We harvested the D.catenatum from 3 test plots, each of which was 1 m2. Spacing per plant was 10 cm×15 cm. About 200 plants were used for the experiment, and only one biennial stem per plant were collected. We didn’t count the exact number of plants, but the total weight of fresh leaves were recorded——1990 g.

Lines 334-339: is this a method developed in your lab? If not, please add reference 

Answer: Yes, this is a method developed in our lab and reported before. A reference has been added in this paper [9].

Lines 342-351: is this a method developed in your lab? If not, please add reference 

Answer: The references have been added in this paper [47, 48].

Is the antioxidant activity evaluated both in the extract prepared for the flavonoids and for the polysaccharides? Please check 

Answer: Yes. The sentence “The solution was used to analyse water soluble polysaccharide content and antioxidant capacity” has been added at the end of 4.5 section. Thank you for your reminder.

Lines 397-412: is this a method developed in your lab? If not, please add reference 

Answer: The references have been added in this paper. They are 48, 53 and 9.

Line 287: “was” instead of “is” 

Answer: The word has been modified as sugested.

Lines 279-307: I suggest you broaden the discussion. You did a great job with so many experiments and results but the discussion is really small compared to the rest of the paper. 

Answer: More discussion has been added in the paper. Please refer directly to the third part of the text. The number of references was changed from 36 to 53 . Their number were 4-6, 8, 21-24, 34-40, 47-19 and 53 in the latest revision.

Round 2

Reviewer 1 Report

The manuscript entitled ‘Effects of Various Processing Methods on the Metabolite Profiles and Antioxidant Capacity of Dendrobium catenatum Leaves’ has been improved following the suggestions. However, some limitations still occur in the revised paper:

-line 42: please, use International System of Units 200,000 mu must be converted in hectares.

-line 44: ‘last 20 years’ should be more appropriate

-line 49: ‘and, in addition,’ should be more appropriate instead of ‘as well as’

-unfortunately, figures 2 and 3 appears with white lines. Please check the quality of image before the publications

-line 179: please specify that are ‘previous results that demonstrated no significant correlation between antioxidant activity and clearance rate of stem polysaccharide extracts

-line 183: please remove italics from ‘which was consistent with previous reports’

-line 184: sorry but for me the explanation for polysaccharides is still unclear as it was exposed. It should be better explain that they might be not present due to the previous extraction with hot water and precipitation

-line 197: ‘RD processes yielded a large number of new, polar metabolites’ is in disagreement with what you stated: ‘No new compounds have been identified in this study, so all the identified compounds were the same found in literature’. However, these results need to be further discussed in comparison with previous findings in literature

-please add the explanation of the choice of flavonoids as compound of interest of your work, basing on literature also in the paper and not only in the author answers.

Author Response

The manuscript entitled ‘Effects of Various Processing Methods on the Metabolite Profiles and Antioxidant Capacity of Dendrobium catenatum Leaves’ has been improved following the suggestions. However, some limitations still occur in the revised paper:

-line 42: please, use International System of Units 200,000 mu must be converted in hectares.

Answer: Thank you very much for your comments and suggestion. The 200,000 mu has been change to 13,000 ha.

-line 44: ‘last 20 years’ should be more appropriate

Answer: ‘last 20 years’ has been modified to ‘since 1990s’ when the breakthrough of artificial cultivation technology, and the sentence has been modified to “It has become one of the fastest development Chinese medicinal materials with the largest market share since 1990s”.

-line 49: ‘and, in addition,’ should be more appropriate instead of ‘as well as’

Answer: The phrase “as well as” has been modified to “and, in addition”.

-unfortunately, figures 2 and 3 appears with white lines. Please check the quality of image before the publications

Answer: From the file opened on my computer, there is no white line in figure 2 or 3. I can provide the original pictures to the editorial department if necessary.

-line 179: please specify that are ‘previous results that demonstrated no significant correlation between antioxidant activity and clearance rate of stem polysaccharide extracts

Answer:  The sentence has been modified to “which was consistent with the previous results that demonstrated no significant correlation between stem polysaccharide content and DPPH· free radical clearance rate”.

-line 183: please remove italics from ‘which was consistent with previous reports’

Answer: In the revised version, this sentence ‘which was consistent with previous reports’ is located in line 226~227, and the format of   has been  corrected.

-line 184: sorry but for me the explanation for polysaccharides is still unclear as it was exposed. It should be better explain that they might be not present due to the previous extraction with hot water and precipitation

Answer: I apologize for the ambiguity in the previous subsection (2.4.). Now it has been modified as below:

Pearson correlation analysis showed a significant positive relationship between flavonoid content and the antioxidant activities of both DPPH and ABTS scavenging rate in 80% methanol extract (0.445 and 0.829, both P < 0.01). However, a significant negative correlation was present between water-soluble polysaccharide content and ABTS+ scavenging ability in water extract (-0.330, P = 0.040). In addition, there was a negative correlation between flavonoid content and water-soluble polysaccharide content (-0.548, P = 0.000). Leaves that had been extracted with water showed no relationship between water-soluble polysaccharide content and DPPH scavenging ability (P = 0.176), which was consistent with the previous results that demonstrated no significant correlation between stem polysaccharide content and DPPH free radical clearance rate [21]. These results indicated that flavonoids were the major antioxidants in the leaves of D. catenatum, which was consistent with previous reports [9,21,22].

-line 197: ‘RD processes yielded a large number of new, polar metabolites’ is in disagreement with what you stated: ‘No new compounds have been identified in this study, so all the identified compounds were the same found in literature’. However, these results need to be further discussed in comparison with previous findings in literature

Answer: Yes, you're right. It's not that new compounds were being produced during D and RD processes. Actually, it's that the content of macropolar metabolites with the rention times of 2~11 min were obviously increased. So the sentence has been modified as below:

As shown in Figure 4B, the height of D and RD samples’ chromatographic peaks at 2~11 min was obviously higher than others, which indicated that different drying processes affected the content of macropolar metabolites.

-please add the explanation of the choice of flavonoids as compound of interest of your work, basing on literature also in the paper and not only in the author answers.

Answer: The explanation has been added in line 77~78: “Since flavonoids and polysaccharides are the main components in leaves [7,9,21], their content was determined by colorimetric method for each sample.”

References

7. Liu, Z.P.; Xu, C.X.; Liu, J.J.; Si, J.P.; Zhang, X.F.; Wu, L.S. Study on accumulation of polysaccharides and alcohol-soluble extracts contents of Dendrobium officinale leaves. China Journal of Chinese Materia Medica 2015, 40, 2314.

9. Zhang, Y.; Zhang, L.; Liu, J.; Liang, J.; Si, J.; Wu, S. Dendrobium officinale leaves as a new antioxidant source. Journal of Functional Foods 2017, 37, 400–415.

21. Zhou, G.; Lv, G. Comparative studies on scavenging DPPH free radicals activity of flavone C-glycosides from different parts of Dendrobium officinale. China journal of Chinese materia medica 2012, 37, 1536-1540.

Reviewer 2 Report

Great job! I think your article is now ready for publication

Author Response

Thank you very much.